# Imbalance-aware loss functions improve medical image classification

**Daniel Scholz**[1,2]                                                    DANIEL.SCHOLZ@MRI.TUM.DE
[1] *Department of Neuroradiology, Technical University of Munich*
[2] *Institute for Artificial Intelligence and Informatics in Medicine, Technical University of Munich*

**Ayhan Can Erdur**[2,3]                                                    CAN.ERDUR@TUM.DE
[3] *Department of Radiation Oncology, Technical University of Munich*

**Josef Buchner**[3]                                                    J.BUCHNER@TUM.DE
**Jan C. Peeken**[3]                                                    JAN.PEEKEN@TUM.DE
**Daniel Rueckert**[*2]                                                    DANIEL.RUECKERT@TUM.DE
**Benedikt Wiestler**[*1]                                                    B.WIESTLER@TUM.DE

**Editors:** Accepted for publication at MIDL 2024

## Abstract

Deep learning models offer unprecedented opportunities for diagnosis, prognosis, and treatment planning. However, conventional deep learning pipelines often encounter challenges in learning unbiased classifiers within imbalanced data settings, frequently exhibiting bias towards minority classes. In this study, we aim to improve medical image classification by effectively addressing class imbalance. To this end, we employ differentiable loss functions derived from classification metrics commonly used in imbalanced data settings: Matthews correlation coefficient (MCC) and the F1 score. We explore the efficacy of these loss functions both independently and in combination with cross-entropy loss and various batch sampling strategies on diverse medical datasets of 2D fundoscopy and 3D magnetic resonance images. Our findings demonstrate that, compared to conventional loss functions, we achieve notable improvements in overall classification performance, with increases of up to +12% in balanced accuracy and up to +51% in class-wise F1 score for minority classes when utilizing cross-entropy coupled with metrics-derived loss. Additionally, we conduct feature visualization to gain insights into the behavior of these features during training with imbalance-aware loss functions. Our visualization reveals a more pronounced clustering of minority classes in the feature space, consistent with our classification results. Our results underscore the effectiveness of combining cross-entropy loss with class-imbalance-aware loss functions in training more accurate classifiers, particularly for minority classes.

**Keywords:** Class imbalance, Deep learning, Loss Function, Unbiased Classifier

## 1. Introduction

Deep learning techniques have revolutionized medical image analysis by providing powerful tools for tasks such as diagnosis, prognosis, and treatment planning (Litjens et al., 2017; Oren et al., 2020; Pinto-Coelho, 2023). However, one significant challenge that persists in training deep learning models for medical image analysis is the presence of imbalanced data (Mazurowski et al., 2008; Buda et al., 2018; Johnson and Khoshgoftaar, 2019). In many

---

* Contributed equally as senior authors

medical datasets, the distribution of classes is often skewed, with certain classes representing the minority while others dominate. This class imbalance poses a substantial hurdle for conventional deep learning pipelines, as they tend to prioritize learning the majority classes at the expense of the minority ones. Consequently, models trained on imbalanced data may exhibit biased predictions, leading to suboptimal performance, particularly for the underrepresented classes critical for accurate diagnosis or prognosis. (Cluceru et al., 2022; Foltyn-Dumitru et al., 2023). The resulting bias against minority classes can have potentially grave consequences for patients when utilizing detection and diagnosis systems based on such biased deep-learning classifiers. This problem is further aggravated since commonly used metrics, such as unbalanced accuracy, convey overly optimistic results in imbalanced classification settings (Haixiang et al., 2017). More appropriate metrics for evaluating classification performance in imbalanced data settings, such as the Matthews correlation coefficient (MCC)(Matthews, 1975; Gorodkin, 2004) and F1 score, have been proposed.

Addressing class imbalance is crucial for developing robust and reliable deep learning models that can generalize well across diverse medical imaging datasets and yield clinically relevant insights. In this study, we focus on mitigating the effects of class imbalance in medical image classification tasks using novel approaches derived from imbalance-aware loss functions, aiming to improve the overall performance and equity across all classes. To this end, we comprehensively compare and analyze class-imbalance-aware loss functions in combination with and against established loss functions in two challenging datasets with imbalanced class distributions. In summary, our contributions are as follows:

1. We investigate different loss formulations and introduce **new combinations of class-imbalance-aware losses** by integrating the MCC and F1 score with a cross-entropy loss.

2. To this end, we **comprehensively compare** different strategies to address class imbalance, namely, class-imbalance-aware loss functions in combination with over-sampling and per-sample weighting, and established loss functions.

3. We experimentally show that **class-imbalance-aware loss functions increase the performance** of challenging classification tasks on diverse medical imaging datasets, as well as the **discernability of class representations**.

## 2. Related Work

### 2.1. Overcoming class imbalance

Considering how relevant a challenge class imbalance is for training clinically applicable deep learning models, several strategies have been developed to overcome this challenge. These can be broadly grouped into (i) sampling-based, (ii) loss-based, and (iii) synthesis-based approaches.

*Sampling-based approaches* typically aim to oversample the minority class(es) or adjust loss weights. While easy to implement and computationally cheap, this can lead to the model severely overfitting the few samples available in the minority classes (Zheng et al., 2015).

Alternatively, specific *loss functions* such as focal loss (Lin et al., 2017) may be used. Focal loss over-proportionally decreases the loss of an easy sample compared to difficult ones. Since the samples in the minority class are potentially more challenging to classify due to their low prevalence, the model is penalized strongly for misclassifying them. However, the focal formulation does not directly address the class imbalance but rather its "side effect" that minority samples are typically more challenging to classify.

Another strategy to tackle class imbalance is creating *synthetic examples*. Earlier examples of such techniques include the Synthetic Minority Over-sampling Technique (SMOTE) (Chawla et al., 2002). In SMOTE, the minority class is over-sampled by taking each sample and introducing synthetic examples along the line segments joining any/all of the $k$ minority class nearest neighbors. Linear interpolation in image space, however, rarely gives sensible synthetic samples. Hence, generative models, such as generative adversarial networks (GANs) (Goodfellow et al., 2014; Qasim et al., 2020; Li et al., 2023) or diffusion models (Qin et al., 2023; Dhariwal and Nichol, 2021) have been developed to create realistic examples to supplement the minority classes. While these are powerful approaches, they are also computationally expensive and require considerable effort to develop generative models that produce realistic, helpful minority class examples, which is again challenging. Extending them to different tasks and classes usually requires extensive re-training.

## 3. Method

### 3.1. Class-imbalance-aware loss functions

To mitigate the imbalanced data issue in medical imaging datasets, imbalance-aware loss functions emerge to enhance the performance of minority classes. These loss functions generally assign larger loss values to misclassified instances of these less prevalent classes. This adaptation serves to rectify the disparity in their impact on the overall loss calculation. We compare the focal loss with two loss functions derived from the MCC and the F1 score.

#### 3.1.1. FOCAL LOSS

Focal Loss (Lin et al., 2017) is an often-used loss function for imbalanced deep-learning classification problems. Difficult-to-classify examples often stem from minority classes. These examples are often predicted with low confidence, yielding higher loss values. Hence, the deep learning model is incentivized to optimize for all classes equally. The loss function is given as

$$\mathcal{L}_{\text{focal}}(p_t) = -(1 - p_t)^\gamma \log(p_t) \tag{1}$$

where the exponent $\gamma$ determines the strength of penalization for samples of class $t$ with predicted probability $p_t$.

#### 3.1.2. SOFT F1 LOSS

The F1 score is a valuable metric for assessing classification performance since it summarizes precision and recall into a single number through the harmonic mean. By macro-averaging the F1 score for all classes, we obtain a balanced assessment of the classifier. We leverage

this property by deriving a negative differentiable F1 score as a loss function. To this end, we use differentiable true positives ($TP$), false positives ($FP$), and false negatives ($FN$):

$$TP = \sum_{i \in I} y_i \cdot \hat{y}_i; \quad FP = \sum_{i \in I} (1 - y_i) \cdot \hat{y}_i; \quad FN = \sum_{i \in I} y_i \cdot (1 - \hat{y}_i) \tag{2}$$

where $y_i$ is the label and $\hat{y}_i$ is the prediction for index $i$. The precision, recall, and F1 score are defined as:

$$\text{precision} = \frac{TP}{TP + FP}; \quad \text{recall} = \frac{TP}{TP + FN}$$
$$F1 = \frac{2 \cdot \text{precision} \cdot \text{recall}}{\text{precision} + \text{recall}} \tag{3}$$

We define the corresponding loss function as $\mathcal{L}_{F1} = 1 - F1_{\text{soft}}$, where $F1_{\text{soft}}$ is the macro average of the F1 score for each class using differentiable (*soft*) $TP$, $FP$, and $FN$.

### 3.1.3. Soft MCC Loss

Matthew's correlation coefficient (MCC) (Matthews, 1975) is a metric that encompasses all four entries of the confusion matrix, namely $TP$, $TN$, $FP$, and $FN$, into a single value in the binary classification case. It has been argued that the MCC is superior to many other metrics, such as accuracy, F1 score, and the receiver operating characteristic (ROC) area under the curve (AUC) (Chicco and Jurman, 2020; Chicco et al., 2021b,a; Chicco and Jurman, 2023), because of the normalization term accounting for class imbalance. We define (*soft*) $TP$, $FP$, and $FN$ as above and additionally calculate true negative ($TN$):

$$TN = \sum_{i \in I} (1 - y_i) \cdot \hat{y}_i \tag{4}$$

From these definitions, $MCC_{soft}$ is defined as:

$$MCC_{soft} = \frac{(TP \cdot TN) - (FP \cdot FN)}{\sqrt{(TP + FP)(TP + FN)(TN + FP)(TN + FN)}} \tag{5}$$

The loss formulation is given as: $\mathcal{L}_{MCC} = 1 - MCC_{soft}$ (Abhishek and Hamarneh, 2021).

### 3.1.4. Combined loss functions

The established cross-entropy loss has desirable theoretical properties. The imbalance-aware losses presented might focus too heavily on the minority class, leading to an unwanted decrease in performance for the majority class. Hence, we also evaluate weighted sums of the F1 and the MCC loss with the cross-entropy loss with equal weights for each loss term.

## 3.2. Addressing class imbalance with sampling and weighting

In balanced data scenarios, the samples in a batch are drawn uniformly, i.e., with the same probability, to obtain an equal uniform distribution of each class in a batch.

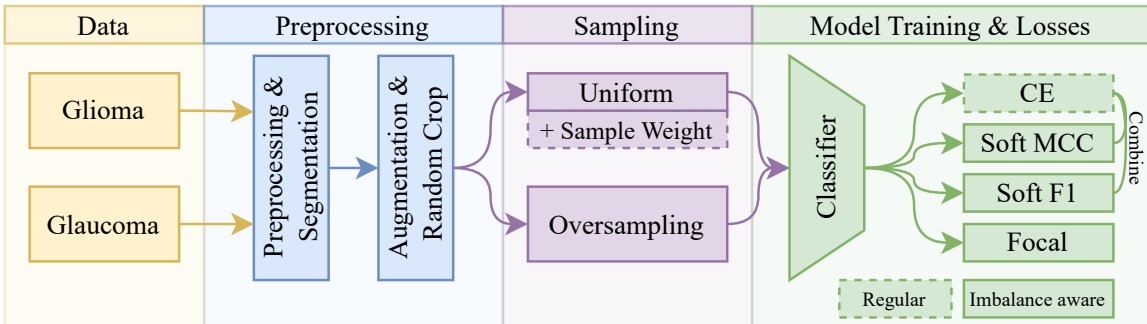

Figure 1: Overview of our study design. We systematically investigate combinations of different sampling strategies, sample weightings, and loss functions (including class-imbalance-aware loss functions).

**Oversampling** One measure to combat class imbalance in deep learning is oversampling the minority class(es) to obtain a more uniform distribution than the original distribution in the dataset. We implement a stratified oversampling technique, i.e., we allocate equal portions of a batch to each class corresponding to equal clinical relevance of each type.

**Sample weighting** Instead of oversampling the minority classes, we can assign higher importance to minority class samples by scaling the influence of a sample according to the prevalence of its class $c \in C$ in the loss calculation:

$$L = \frac{1}{n} \sum_i w_c^{(i)} L^{(i)} \tag{6}$$

We choose a normalized inverse frequency (Sparck Jones, 1972) scaling for each sample.

## 4. Experiment Setup

### 4.1. Experiment pipeline

An overview of our experimental design is shown in Figure 1. We conduct our experiments on two diverse and imbalanced datasets. We apply standard preprocessing and augmentation techniques. Two sampling strategies are used to train a ResNet classifier (He et al., 2016). We compare six loss functions, cross-entropy (CE), focal loss, soft F1, soft MCC, and CE with MCC and F1. For all experiments using the cross-entropy-loss we also compare sample weighting while sampling uniformly.

We share our dataset configurations and the code used for our study at `https://github.com/daniel-scholz/address-class-imbalance`. For further details of our experiments implementation, refer to Appendix B.

### 4.2. Datasets

To allow the reproduction of our results, our study only uses publicly available datasets.

**Glioma** The first dataset comprises 3D MR images (T1w -/+ contrast, T2w, FLAIR) from large public datasets of adult patients with newly diagnosed gliomas, namely UCSF-PDGM (Calabrese et al., 2022), EGD (van der Voort et al., 2021), and TCGA (Bakas et al., 2017). Besides having all four imaging sequences outlined above available, we require biomarker testing for *IDH* mutation and 1p/19q status in order to classify samples according to the 2021 WHO classification of brain tumors into (a) *IDH* wildtype glioblastoma, (b) *IDH* mutant and 1p/19q intact astrocytoma and (c) *IDH* mutant and 1p/19q codeleted oligodendroglioma (Wen and Packer, 2021). In total, our dataset contains pre-operative MRIs of 1174 patients. The prevalence of glioblastoma (∼80%) in comparison to oligodendroglioma (∼8%) and astrocytoma (∼12%) is striking and consistent throughout all the available datasets, mirroring the real-world distribution. A visualization of the class distributions is shown in the Appendix C, Figure 5. We hold the TCGA dataset out for testing and use the remaining data for training. For additional robustness analysis, we run each experiment with four different network initializations.

**Glaucoma** The second dataset consists of 1542 individual 2D RGB fundus photographs, of which 786 are healthy controls, 289 photographs show early glaucoma, and 467 are from advanced glaucoma patients (Ahn et al., 2018). We randomly split the dataset into training ($\frac{3}{4}$) and testing ($\frac{1}{4}$) data, stratified by class: {no, early, advanced} glaucoma.

## 5. Results

### 5.1. Global classification results

Our main results on the test sets are shown in Table 1 (and Appendix A, Table 3, 4, and 5). The baseline model (CE loss, uniform sampling) shows reasonable performance, which improves when adding sample weights or oversampling the minority classes for both datasets. The most considerable improvement over the baseline qua balanced accuracy is achieved when using the CE + soft F1 loss combination and oversampling, with a relative improvement of 12.7% on the glioma dataset. We also observe smaller standard deviations for the loss combinations compared to the single class-imbalance-aware losses, F1 and MCC, indicating better training stability for the combination. We explore this further in an ablation study (Appendix A, Table 6) for small batch size regimes. The performance on the glaucoma dataset also improves most with the CE + F1 loss combination and oversampling (+10.5%).

### 5.2. Per-class analysis

In addition, we perform a per-class analysis of our methods using the class-wise F1 score, which balances precision and recall (Table 2). The baseline training setup yields a classifier biased towards the majority class (glioblastoma / no glaucoma) while performing poorly on the minority classes. The overall improvements in classification performance can be directly traced to improved minority class performance, since majority class performance stays constant across almost all experiments. The CE + F1 loss tremendously improves classification performance on the astrocytoma minority class (+20.3%). The largest improvement when using CE + F1 with oversampling loss is observed on the least prevalent early glaucoma (+51.3%). However, we also observe that using only a class imbalance-aware loss sometimes yields classifiers entirely ignoring one class (e.g., F1 loss in the glaucoma).

Table 1: Multi-class classification results for six different loss functions, two sampling methods, and two different medical imaging datasets in terms of **balanced accuracy**($\uparrow$). Results for the glioma dataset are mean$_{\pm \text{std}}$.

| Dataset | Sampling | CE | CE + F1 | CE + MCC | F1 | Focal | MCC |
|---|---|---|---|---|---|---|---|
| Glioma | Uniform | $0.55_{\pm0.03}$ | $0.57_{\pm0.02}$ | $0.59_{\pm0.03}$ | $0.51_{\pm0.12}$ | $0.57_{\pm0.04}$ | $0.39_{\pm0.10}$ |
| | + Weights | $0.59_{\pm0.03}$ | $0.60_{\pm0.02}$ | $\mathbf{0.62}_{\pm0.03}$ | - | - | - |
| | Oversampling | $0.61_{\pm0.01}$ | $\mathbf{0.62}_{\pm0.01}$ | $0.59_{\pm0.04}$ | $0.60_{\pm0.03}$ | $0.61_{\pm0.04}$ | $0.59_{\pm0.03}$ |
| Glaucoma | Uniform | 0.69 | 0.70 | 0.72 | 0.59 | 0.70 | 0.59 |
| | + Weights | 0.74 | 0.76 | 0.75 | - | - | - |
| | Oversampling | 0.75 | **0.77** | 0.76 | 0.58 | 0.74 | 0.50 |

Table 2: Comparison for different class performances in terms of **F1 score** ($\uparrow$). The majority class performance has little variance while the minority class performance improves tremendously for class imbalance aware losses and cross-entropy with oversampling (Astro: astrocytoma, GB: glioblastoma, Oligo: oligodendroglioma).

| Loss | Sampling | Glaucoma | | | Glioma | | |
|---|---|---|---|---|---|---|---|
| | | No | Early | Advanced | GB | Astro | Oligo |
| CE | Uniform | **0.85** | 0.37 | 0.83 | $0.87_{\pm0.00}$ | $0.54_{\pm0.06}$ | $0.30_{\pm0.06}$ |
| | + Weights | 0.80 | 0.54 | 0.81 | $0.88_{\pm0.01}$ | $0.63_{\pm0.02}$ | $0.30_{\pm0.06}$ |
| | Oversampling | 0.79 | 0.55 | 0.79 | $0.88_{\pm0.01}$ | $0.63_{\pm0.01}$ | $0.34_{\pm0.04}$ |
| CE + F1 | Uniform | **0.85** | 0.42 | 0.81 | $0.87_{\pm0.01}$ | $0.60_{\pm0.04}$ | $0.30_{\pm0.05}$ |
| | + Weights | 0.80 | **0.56** | 0.84 | $0.88_{\pm0.01}$ | $0.60_{\pm0.02}$ | $0.34_{\pm0.03}$ |
| | Oversampling | 0.81 | 0.54 | **0.85** | $\mathbf{0.89}_{\pm0.00}$ | $0.65_{\pm0.03}$ | $0.34_{\pm0.00}$ |
| CE + MCC | Uniform | **0.85** | 0.44 | 0.83 | $0.87_{\pm0.01}$ | $0.60_{\pm0.03}$ | $\mathbf{0.38}_{\pm0.11}$ |
| | + Weights | 0.78 | 0.55 | 0.83 | $\mathbf{0.89}_{\pm0.01}$ | $0.62_{\pm0.06}$ | $\mathbf{0.38}_{\pm0.04}$ |
| | Oversampling | 0.82 | 0.53 | 0.83 | $0.88_{\pm0.01}$ | $0.59_{\pm0.06}$ | $0.32_{\pm0.07}$ |
| F1 | Uniform | 0.81 | 0.00 | 0.74 | $0.85_{\pm0.04}$ | $0.52_{\pm0.35}$ | $0.07_{\pm0.15}$ |
| | Oversampling | 0.81 | 0.00 | 0.70 | $0.89_{\pm0.00}$ | $0.66_{\pm0.02}$ | $0.25_{\pm0.07}$ |
| Focal | Uniform | 0.84 | 0.44 | 0.83 | $0.87_{\pm0.02}$ | $0.63_{\pm0.03}$ | $0.25_{\pm0.11}$ |
| | Oversampling | 0.80 | 0.52 | 0.81 | $0.87_{\pm0.02}$ | $0.62_{\pm0.08}$ | $0.35_{\pm0.04}$ |
| MCC | Uniform | 0.82 | 0.05 | 0.73 | $0.81_{\pm0.04}$ | $0.17_{\pm0.33}$ | $0.00_{\pm0.00}$ |
| | Oversampling | 0.00 | 0.32 | 0.72 | $0.88_{\pm0.01}$ | $\mathbf{0.70}_{\pm0.01}$ | $0.15_{\pm0.14}$ |

## 5.3. Visual feature space analysis

To investigate the learned representations, we plot the features of the last ResNet layer for the glioma dataset. We use the popular t-distributed stochastic neighbor embedding (tSNE) (Van der Maaten and Hinton, 2008) to project the 256-dimensional feature vectors to 2D for visualization purposes (see Figure 2). We observe that representations of the oligodendroglioma are often poorly clustered in this feature space, corresponding to the inferior performance in this class observed in Table 2. CE + MCC loss with uniform sampling

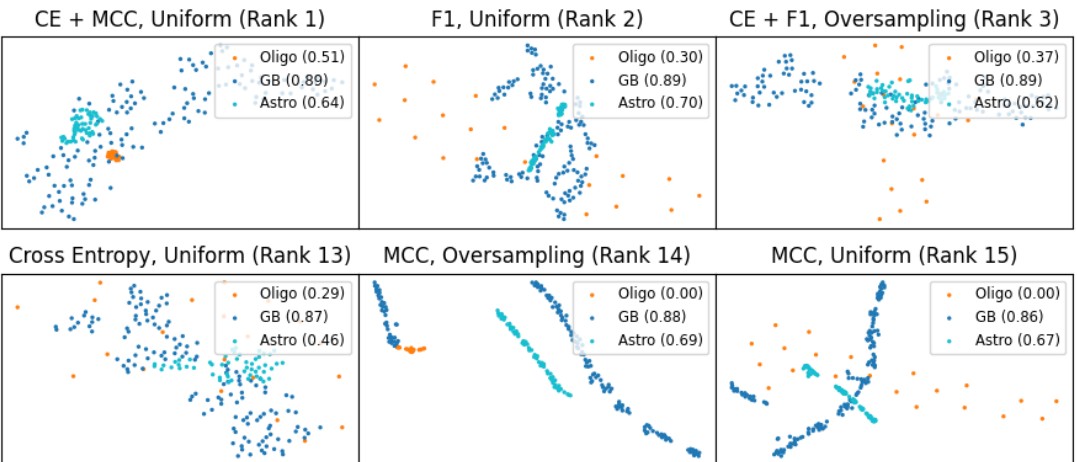

Figure 2: tSNE visualization of the feature representations before the last layer of the best and worst models (1st run) according to the F1 score (↑) with class-wise F1 scores.

shows a better clustering of the oligodendroglioma features compared to the baseline or the MCC-only loss.

## 6. Discussion and Conclusion

Class imbalance-aware loss functions relevantly enhance classification performance, primarily by improving recognition of minority classes. Notably, the most substantial enhancements are observed when combining F1 or MCC loss with standard cross-entropy loss. In particular, this combination seems to stabilize training. This points to similarities between classification and segmentation methodologies, exemplified in frameworks like nnUNet (Isensee et al., 2021), where the combination of Dice (which essentially is the F1 score) and cross-entropy is found to perform best, and underscores the effectiveness of such hybrid approaches. Further, it's worth noting that when employing only class imbalance-aware loss functions, there can be instances where certain classes may be somewhat neglected, a scenario not encountered in the combined approach. In summary, the demonstrated efficacy of class imbalance-aware loss functions, alongside their ease of implementation, computational efficiency, and adaptability across various medical imaging tasks, highlights their potential impact on advancing clinical applications and enhancing the accuracy and reliability of deep learning-based diagnostics in real-world healthcare settings. These properties call for future studies exploring more scenarios in medical image classification with class-imbalance-aware loss functions such as different network architectures or time-series data. Ultimately, we show that integrating loss functions derived from popular metrics such as the F1 score and the MCC with standard cross-entropy loss results in more robust classifiers, with particular benefits for the minority class, thus underscoring its clinical relevance.

## Acknowledgments

This study was supported by the DFG, grant #504320104.

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

# Appendix A. Further results

## A.1. Additional metrics

Table 3: Multi-class classification results for two different medical imaging datasets in terms of **F1 score**(↑). Results for the glioma dataset are mean$_{\pm\text{std}}$.

| Dataset | Sampling | CE | CE + F1 | CE + MCC | F1 | Focal | MCC |
|---|---|---|---|---|---|---|---|
| Glioma | Uniform | $0.57_{\pm0.03}$ | $0.59_{\pm0.02}$ | $0.62_{\pm0.04}$ | $0.48_{\pm0.16}$ | $0.59_{\pm0.05}$ | $0.32_{\pm0.12}$ |
| | + Weights | $0.60_{\pm0.03}$ | $0.61_{\pm0.02}$ | $\mathbf{0.63}_{\pm0.02}$ | - | - | - |
| | Oversampling | $0.62_{\pm0.01}$ | $0.62_{\pm0.01}$ | $0.59_{\pm0.04}$ | $0.60_{\pm0.03}$ | $0.61_{\pm0.04}$ | $0.58_{\pm0.05}$ |
| Glaucoma | Uniform | 0.71 | **0.73** | **0.73** | 0.52 | 0.70 | 0.53 |
| | + Weights | 0.68 | 0.69 | 0.71 | - | - | - |
| | Oversampling | 0.72 | **0.73** | 0.72 | 0.50 | 0.71 | 0.35 |

Table 4: Multi-class classification results in terms of **macro-averaged Area under the ROC Curve**(↑). Results for the glioma dataset are mean$_{\pm\text{std}}$.

| Dataset | Sampling | CE | CE + F1 | CE + MCC | F1 | Focal | MCC |
|---|---|---|---|---|---|---|---|
| Glioma | Uniform | $0.80_{\pm0.02}$ | $0.81_{\pm0.01}$ | $0.81_{\pm0.02}$ | $0.74_{\pm0.12}$ | $0.79_{\pm0.03}$ | $0.57_{\pm0.13}$ |
| | + Weights | $0.80_{\pm0.01}$ | $0.81_{\pm0.02}$ | $\mathbf{0.82}_{\pm0.01}$ | - | - | - |
| | Oversampling | $0.81_{\pm0.01}$ | $\mathbf{0.82}_{\pm0.02}$ | $0.81_{\pm0.02}$ | $0.77_{\pm0.03}$ | $0.79_{\pm0.02}$ | $0.76_{\pm0.01}$ |
| Glaucoma | Uniform | 0.90 | 0.89 | 0.90 | 0.75 | 0.89 | 0.78 |
| | + Weights | 0.90 | 0.90 | 0.90 | - | - | - |
| | Oversampling | 0.90 | 0.90 | 0.89 | 0.79 | 0.88 | 0.57 |

Table 5: Multi-class classification results for different medical imaging datasets in terms of **Matthews correlation coefficient (MCC)**(↑). Results for the glioma dataset are mean$_{\pm\text{std}}$.

| Dataset | Sampling | CE | CE + F1 | CE + MCC | F1 | Focal | MCC |
|---|---|---|---|---|---|---|---|
| Glioma | Uniform | $0.47_{\pm0.03}$ | $0.51_{\pm0.03}$ | $0.52_{\pm0.02}$ | $0.41_{\pm0.27}$ | $0.51_{\pm0.05}$ | $0.13_{\pm0.25}$ |
| | + Weights | $0.52_{\pm0.03}$ | $0.52_{\pm0.02}$ | $0.54_{\pm0.05}$ | - | - | - |
| | Oversampling | $0.53_{\pm0.02}$ | $\mathbf{0.55}_{\pm0.01}$ | $0.50_{\pm0.04}$ | $0.54_{\pm0.02}$ | $0.52_{\pm0.05}$ | $0.56_{\pm0.03}$ |
| Glaucoma | Uniform | 0.63 | 0.64 | **0.65** | 0.53 | 0.63 | 0.53 |
| | + Weights | 0.62 | 0.64 | 0.63 | - | - | - |
| | Oversampling | 0.61 | 0.64 | 0.64 | 0.52 | 0.61 | 0.27 |

## A.2. Ablation: Small batch size regimes

To better understand the benefits of combining a class-imbalance aware loss function with cross-entropy loss, we performed an experiment in a small batch size regime, which is commonly found in 3D medical image analysis. The results from three independent runs are shown in Table 6. Noteworthy, we observe clearly lower standard deviations when combining soft MCC with cross-entropy loss, indicating a stabilizing effect of combining both losses.

Table 6: Comparison of soft MCC loss with and without additional cross-entropy loss in small batch size regimes (batch size = 10).

| Loss | Sampling | Balanced Accuracy | MCC | AUC |
|---|---|---|---|---|
| MCC | Uniform | $0.33_{\pm 0.00}$ | $0.00_{\pm 0.00}$ | $0.50_{\pm 0.00}$ |
|  | Oversampling | $0.51_{\pm 0.12}$ | $0.39_{\pm 0.28}$ | $0.66_{\pm 0.12}$ |
| CE + MCC | Uniform | $\mathbf{0.60}_{\pm 0.01}$ | $\mathbf{0.53}_{\pm 0.01}$ | $\mathbf{0.81}_{\pm 0.01}$ |
|  | Oversampling | $\mathbf{0.60}_{\pm 0.03}$ | $0.51_{\pm 0.03}$ | $0.81_{\pm 0.01}$ |

# Appendix B. Implementation details

## B.1. Glioma dataset

### B.1.1. Image preprocessing and segmentation

All images were preprocessed and segmented using the publicly available BraTS Toolkit (Kofler et al., 2020). After tumor segmentation, images are [0;1] normalized within the brainmask. A $96^3$ patch, centered around the center of mass of the tumor mask, is cropped from the image.

### B.1.2. Data augmentation

We incorporate a range of randomized image intensity and geometry augmentations with a probability of 0.5. The set of intensity-changing augmentations consists of randomly adjusting gamma values within the range of 0.5 to 1.5 and Gaussian blur, with a standard deviation varying randomly between 0 and 1.5. In our geometric augmentations, we randomly flip along the sagittal, coronal, or axial planes and randomly crop with a randomized center, selecting a $64^3$ cube within an already cropped tumor region to introduce more variability in the tumor's positioning.

## B.2. Glaucoma dataset

### B.2.1. Image preprocessing

All images were resized to $240 \times 240$ px and [0;1] normalized.

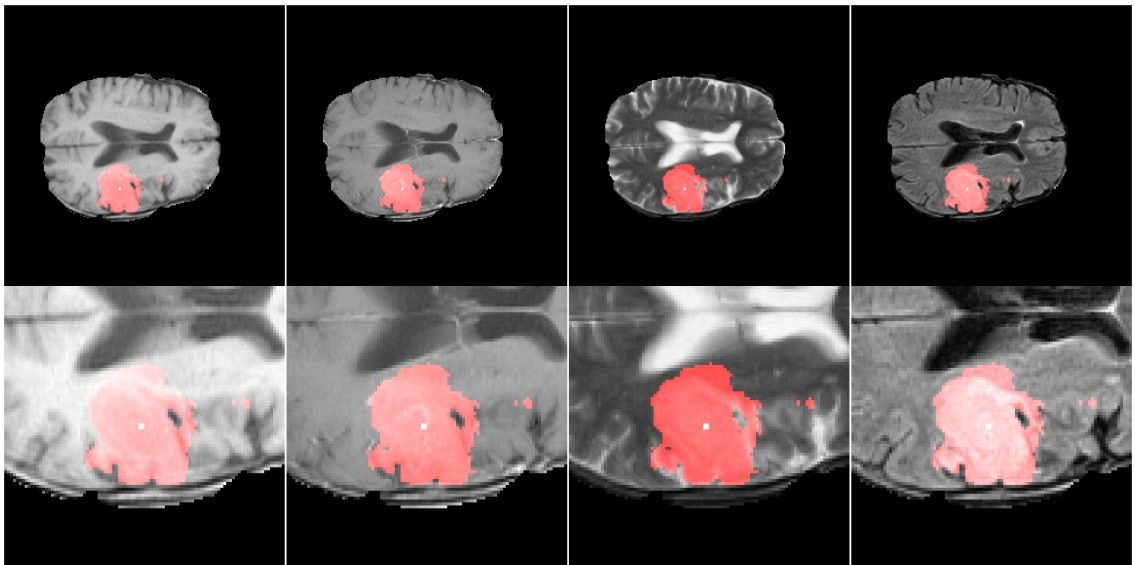

Figure 3: Visualization of the four input sequences available in our dataset. The top row shows entire slices (with the tumor segmentation overlaid in red), and the bottom row shows the crops used for model training.

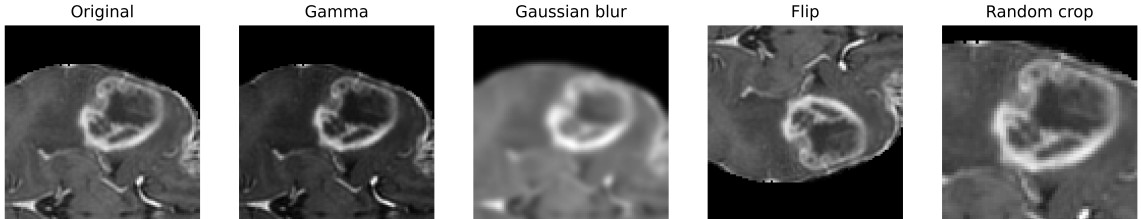

Figure 4: Visualization of the data augmentations used to train our classifier.

### B.2.2. DATA AUGMENTATION

For the glaucoma dataset, we also include a range of randomized image intensity and geometry augmentations with a probability of 0.5 each: The set of intensity-changing augmentations consists of randomly adjusting gamma values within the range of 0.5 to 1.5 and contrast adjustment with a gain randomly selected between [5.,10.]. In our geometric augmentations, we randomly flip along the horizontal or vertical axis.

## B.3. Model training

Our classifier is a ResNet34 (He et al., 2016) architecture composed of [3;4;6;3] residual blocks, adapted to 3D. We implement the neural network and training using TensorFlow 2.14 (Martín Abadi et al., 2015), the gamma augmentations with Scikit-image 0.22.0 (van der Walt et al., 2014), and the Gaussian filtering with Scipy 1.11.3 (Virtanen et al., 2020). We use Adam optimizer (Kingma and Ba, 2015), with parameters $\beta_1 = 0.9$, $\beta_2 = 0.999$, a

learning rate of 1e-3, and a batch size of 50. We also employ a cosine annealing learning rate scheduler (Loshchilov and Hutter, 2016), with a maximum of 250 epochs without warm-up.

## Appendix C. Dataset Distributions

For improved visualization of the class imbalance present in the datasets used, we show the class distributions of all datasets in Figure 5. The distribution over the whole Glioma dataset is very similar to the individual sub-datasets, pointing to a skewed real-world distribution of Gliomas.

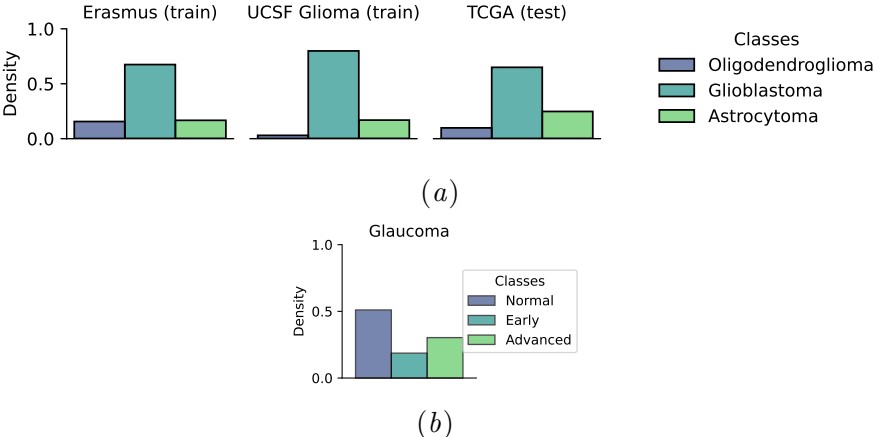

$(a)$

$(b)$

Figure 5: Class distributions for the Glioma and Glaucoma dataset.

