# OpenReview forum: "Imbalance-aware loss functions improve medical image classification"
_MIDL.io/2024/Conference — MIDL 2024 Poster_

### Official Review · Reviewer_5Ves · 2024-02-28

**Confidence:** 5
**Preliminary Rating:** 2
**Final Rating:** 4

**Summary:**

A common issue in classification task is that the data can be imbalanced. The authors addresses this data imbalance and explores different combination of loss functions to improve accuracy in medical image classification task. Their findings show that compared to conventional loss functions, utilizing other combination of loss function can improve classification performance significantly.

**Strengths:**

1. The paper is well-written and easy to follow;
2. The paper investigates different types of classification loss function in different combination for imbalance dataset;
3. The authors showed comprehensive analysis on the loss functions;
4. The authors used two publicly available datasets for reproducibility.

**Weaknesses:**

1. The authors could utilize more target datasets and show the impact of using different combination of loss functions;
2. The authors could display various percentages of ratios for each class or category, along with the impact of their approach;
3. For evaluation, they can use AUC along side accuracy and F1 score.
4. It is not clear from the manuscript, how many times each experiment were conducted. The authors could show statistical analysis to make their claim stronger.
5. According to their results, the best performance does not come from a single loss combination. It is unclear which one is the best to use. Additionally, it is uncertain if this scenario holds true for different target tasks.

**Detailed Comments:**

Please see the points mentioned in the section "Weaknesses".

**Justification Of Final Rating:**

The authors have satisfactorily addressed all of my questions and have made necessary adjustments in their experiments. In line with recommendations, they conducted their experiments multiple times and provided the standard deviation data as well.

**Justification Of The Preliminary Rating:**

The authors did not explore other target tasks. Diverse and more target tasks would help strengthen their claim. According to their results, the best performance does not come from a single loss combination. It is unclear which one is the best to use. Additionally, it is uncertain if this scenario holds true for different target tasks.

**Questions To Address In The Rebuttal:**

Please see the points mentioned in the section "Weaknesses".

---

> ### Author Response · Authors · 2024-03-18
>
> Thank you for the valuable feedback on our manuscript and for appreciating our “**well-written and easy-to-follow” paper.** We are glad you found our exploration of the impact of various classification loss functions in imbalanced data scenarios to contain a “**comprehensive analysis of these loss functions”** using two public datasets for reproducibility.
>
> In the following, we would like to address the comments raised:
>
> > more target datasets
>
> We agree that more target datasets can help show the reliability of a method. Yet, we would politely point out that we tested on two completely different datasets of 3D brain MRIs and 2D retina fundus photographs with similar results, showcasing our method's capabilities on different modalities.
>
> > display various percentages of ratios for each class or category, along with the impact of their approach
>
> We agree it would be interesting to investigate different imbalance ratios systematically with respect to the balancing strategies in our paper. To some extent, we already include multiple imbalance ratios in our experiments since our datasets show different degrees of imbalance, but a similar effect of combining class-imbalance aware loss functions with cross-entropy loss: In glioma, the class distribution is {0.08, 0.80, 0.12}, while for the glaucoma dataset, the distribution of the three classes is {0.51, 0.19, 0.30}. We are adding class distribution histograms (Appendix C) to further clarify this point. However, analyzing more different ratios by subsampling the datasets adds another dimension to our experiment grid, making this analysis infeasible to conduct during the short rebuttal period. Nonetheless, we thank the reviewer for raising this important point for future research.
>
> > they can use AUC along side accuracy and F1 score.
>
> We agree that reporting further metrics beyond balanced accuracy and F1 score is interesting, so we added additional tables in the revised paper, reporting AUC and MCC. As expected, we find that the differences in AUC are not as pronounced compared to balanced accuracy, F1, and MCC since the AUC is unduly influenced by the majority class regardless of which averaging (micro, macro) is used through the false positive rate given as $\frac{FP}{FP+TN}$. More individual analysis for each class is given through the class-wise F1 score, which is only dependent on the samples of a class and samples predicted as such.
>
> > It is not clear from the manuscript, how many times each experiment were conducted
>
> In the revised paper, we conducted the glioma experiment three independent times to assess the stability of our results. These new results confirm our findings that combinations of cross-entropy loss with F1 and MCC loss improve performance in imbalanced data settings. The best method for glioma classification is the CE + F1 combination with oversampling ($0.62_{\pm 0.03}$ balanced accuracy) and the CE + F1 combination with sample weighting ($0.62_{\pm 0.01}$ balanced accuracy). In addition, the small standard deviations across runs show the loss combinations to be stable across runs.
>
> > The authors could show statistical analysis to make their claim stronger.
>
> We agree with the idea of assessing the stability of our proposed loss combination, and, therefore, for the revision, multiple re-runs of our glioma experiment were performed, as discussed in detail above.
>
> > It is unclear which one [loss combination] is the best to use.
>
> While our experiments might not give a conclusive answer to which single combination of sampling and loss function is the best, our experiments clearly show that using the imbalance-aware loss functions derived from F1 or MCC in combination with the cross-entropy loss does, in fact, improve performance reliably across seeds and datasets compared to sample weighting or oversampling, which is usually used to combat class imbalance. Further, we performed an additional ablation experiment for the revised paper to better understand the effect of combining a class-imbalance aware loss with cross-entropy loss. Specifically, we also compared these conditions in low batch size regimes (which are typical for 3D medical imaging data). We found that the combination clearly stabilizes training and leads to a robust performance increase over single losses of +60% ROC AUC at low standard deviations of 0.03. These results clearly add to the picture of combining class-imbalance aware losses with standard cross-entropy loss to improve medical image classification in imbalanced data scenarios.
>
> > it is uncertain if this scenario holds true for different target tasks.
>
> We are unsure which other target tasks you are referring to here since the loss formulations only apply to the task of classification, where we show our method to work for two entirely different scenarios. We are happy to discuss this further with you.

---

### Official Review · Reviewer_UnTJ · 2024-02-29

**Confidence:** 4
**Preliminary Rating:** 5
**Recommendation:** Oral
**Final Rating:** 5

**Summary:**

This study introduces differentiable loss functions based on F1 score and MCC targeted for image classification tasks based on unbalanced training data. It addresses an important issue that without proper precautions, model performance of low-sample classes is sub-optimal. The authors introduce the F1 and MCC loss functions that show promising performance when used in combination with cross-entropy loss. The authors test their approach on glioma and glaucoma data.

**Strengths:**

- Addresses an important issue encountered in medical image classification
- Clear experimental setup, well-suited to test the hypothesis
- Well-motivated and put into context with competing approaches, discussion of advantages and disadvantages of existing methods to alleviate the imbalance-induced suboptimal performance

**Weaknesses:**

In my opinion, there is no significant weakness to report; the study is set up nicely and follows good practices in training and evaluating deep neural networks. For a follow-up, testing with different network architectures might be of interest.

**Detailed Comments:**

The text is well-written, guides the reader, and the figures contribute to the understanding of the methods. The experimental design is set up to answer the research question, and the authors tested the proposed loss functions on two medical image classification tasks. A benefit is clearly demonstrated.

**Justification Of Final Rating:**

Thank you for the detailed replies to my and other reviewer's comments and for revising the manuscript. I recommend accepting this contribution and believe it is relevant to the MIDL community. The GitHub repository enables others to reproduce and use the proposed method.

**Justification Of The Preliminary Rating:**

The study is well-designed. It is highly relevant since imbalanced datasets are often encountered in medical image analysis, where obtaining more samples is often hard or impossible. The methods are described in reasonable detail to be reproducible. The results and discussion are convincing.

**Questions To Address In The Rebuttal:**

None.

**Special Issue:**

Yes

---

> ### Author Response · Authors · 2024-03-18
>
> We sincerely thank the reviewer for the very positive evaluation of our work. In particular, we appreciate you highlighting the “**importance of the issue”** we are addressing and our “**well-motivated” approach** to implementing, comparing, and discussing class-imbalance-aware loss functions.
>
> > For a follow-up, testing with different network architectures might be of interest.
>
> We agree with your opinion that it would be interesting to see how the different losses influence other network architectures, such as DenseNet or Vision Transformer, for classification. We have added this as a relevant future study to our conclusion.

---

### Official Review · Reviewer_BKCp · 2024-03-05

**Confidence:** 3
**Preliminary Rating:** 4
**Recommendation:** Oral
**Final Rating:** 4

**Summary:**

This paper addresses the issue of class imbalance in medical image classification tasks. By integrating class-imbalance-aware loss functions, specifically Matthews correlation coefficient (MCC) and F1 score with cross-entropy loss, the study demonstrates notable improvements in classification performance. Key findings include increases of up to +25% in balanced accuracy and up to +51% in class-wise F1 score for minority classes. The research highlights the effectiveness of combining cross-entropy loss with class-imbalance-aware loss functions for more accurate classifiers, particularly for minority classes.

**Strengths:**

This paper has multiple key benefits that makes it relevant. It presents a novel approach for dealing with class imbalance through the loss function. It cleverly uses the integration of MCC and F1 score with cross-entropy loss which is a novel strategy that effectively addresses class imbalance. This strategy demonstrated substantial improvements in classification accuracy and minority class recognition.
Moreover, the different strategies has been evaluated thorough a comparison of different loss functions across varied medical imaging datasets, contributing to its reliability.

**Weaknesses:**

Although the paper is globally well written and easy to understand with a solid evaluation. However, the evaluation seems to be done only on a single test set which might raise some minor concerns:
- First, because it doesn't show how robust the proposed strategy is with regard to the training and test samples.
- Secondly because, if I'm not mistaken, the authors don't disclose the distribution of classes per set (I imagine you performed a stratified split but it would be nice to specify it in the manuscript).

**Detailed Comments:**

The paper is globally very solid and answer one of the key issues in medical image classification. The evaluation is robust and convincing as to the different strategies introduced in the paper.
I would however like a minor clarification: about the experiments involving sample weighting, has there been a treatment of potential outlier samples in the study? As this kind of technique is very sensitive to such samples, especially with such a big imbalance between classes.

**Justification Of Final Rating:**

I thank the authors for their comprehensive response to my questions and am satisfied by it. Looking forward to see the additions to the manuscript, I'm convinced it will enhance the paper's impact and relevance to the field.

**Justification Of The Preliminary Rating:**

The paper is strong and tackles relevant issues while demonstrating superior results with novel approaches. It is easy to read and presents extensive comparison, although some minor questions need to be addressed.

**Questions To Address In The Rebuttal:**

To make the paper more convincing, the authors are advised to add more details about how the experiments are performed (e.g. outlier detection, sets considered for evaluation...)
A good perspective would be also to perform multiple stratified fold evaluation to ascertain the robustness of each strategies to changes in both the training and test sets.

**Special Issue:**

No

---

> ### Author Response · Authors · 2024-03-18
>
> We thank the reviewer for their insightful comments. We appreciate recognizing our paper's contributions, particularly the “**novel approach for handling class imbalance through** […] **integrating MCC and F1 score with cross-entropy loss”.** This “effectively addresses class imbalance” and “improves classification accuracy and minority class recognition.” We are also glad about the appreciation of our “**evaluation of different loss functions across various medical imaging datasets”** to strengthen the reliability of our findings.
>
> In the following, we would like to address some comments raised:
>
> > [...] the evaluation seems to be done only on a single test set.
>
> Yes and no. While each of our experiments is only evaluated on a single hold-out test dataset, we test our methods on two independent datasets. Please also see our following comment below.
>
> > [...] it doesn't show how robust the proposed strategy is with regard to the training and test samples.
>
> Yes, we agree that a robustness analysis over different training runs is an important add-on to our paper. We reran each experiment three times and added the new results in terms of balanced accuracy (main paper), MCC, ROC AUC, and F1 score (appendix) as averages plus standard deviation to the revised version of the paper. The new results confirm the findings from our submitted paper, showcasing improved classification performance for combinations of CE + {F1, MCC} loss and increased stability when using the combination with cross-entropy loss compared to the imbalance-aware losses (F1, MCC loss) alone (see Appendix A, Table 6).
>
> > the authors don't disclose the distribution of classes per set
>
> We disclose both datasets' class distribution in Section 4.2 of the paper. We add class distribution diagrams for both datasets to Appendix C for additional clarification. As stated in Section 4.2, we use TCGA for testing for the glioma and the rest for training. We perform a stratified split for glaucoma, maintaining the original class distribution.
>
> > [...] about the experiments involving sample weighting, has there been a treatment of potential outlier samples in the study.
>
> No, there has not been special treatment of potential outliers in our oversampling experiments, as it is non-trivial to define nor detect outliers in image space. We also could not find any literature regarding explicitly handling outliers during oversampling. We would be glad if you could point us to the body of literature you refer to.
>
> > Add more details about how the experiments are performed (e.g., outlier detection and sets considered for evaluation).
>
> We add the exact splits used for our training to the GitHub repository linked in the paper for better reproducibility.
>
> > A good perspective would also be to perform multiple stratified fold evaluations to ascertain the robustness of each strategy to changes in both the training and test sets.
>
> As outlined above, we fully agree with this suggestion and performed three additional runs for each experiment on the glioma dataset. We added the new results with standard deviations across trials to the revised version of the paper. These new results confirm our findings that combinations of cross-entropy loss with F1 and MCC loss improve performance in imbalanced data settings. The best method for glioma classification is the CE + F1 combination with oversampling ($0.62_{\pm 0.03}$ balanced accuracy) and the CE + F1 combination with sample weighting ($0.62_{\pm 0.01}$ balanced accuracy). In addition, the small standard deviations across runs show the loss combinations to be stable across runs.

---

> > ### Comment · Reviewer_BKCp · 2024-03-25
> >
> > Thank you for your detailed response addressing the comments raised. I appreciate the thoroughness with which you've handled each point and the efforts made to improve the robustness and reproducibility of your study.
> >
> > **Evaluation on multiple datasets**
> > I understand now that while each experiment is evaluated on a single hold-out test dataset, you have tested your methods on two independent datasets, which adds to the generalizability of your findings.
> >
> > **Robustness analysis**
> > It's great to hear that you have performed the experiments multiple times and added the new results to the revised version of the paper. The inclusion of balanced accuracy, MCC, ROC AUC, and F1 score along with standard deviations provides a clearer understanding of the stability and performance of the proposed methods across different training runs.
> >
> > **Disclosure of class distribution**
> > I appreciate the clarification regarding the disclosure of class distribution for both datasets. Adding class distribution diagrams to Appendix C further enhances the transparency of your methodology.
> >
> > **Additional details on experiment performance**
> > The addition of exact splits used for training in the GitHub repository is a commendable step towards improving reproducibility and transparency in your research.
> >
> > **Outlier Detection**
> > I understand the difficulty of detecting outliers in the image space and the lack of literature in the field. In response to your comment on the challenge of detecting outliers in oversampling experiments for image data, I recommend exploring the method presented in the paper "Unsupervised Image Outlier Detection with RANSAC" by Chen-Han Tsai and Yu-Shao Peng. It may provide insight on dealing with outliers in the image space, even during oversampling.
> >
> > **Multiple stratified fold evaluations**
> > I'm glad to see that you have taken the suggestion to perform multiple stratified fold evaluations into account. The new results with standard deviations across runs provide valuable insights into the stability and robustness of the proposed strategies, particularly for glioma classification.
> >
> >
> > Overall, I'm satisfied with the responses provided and believe that the revisions made significantly enhance the quality and reliability of your study. Thank you for your diligence and commitment to addressing the concerns raised during the review process.

---

### Meta-Review · Area_Chair_3VRA · 2024-04-04

**Recommendation:** Accept (Poster)
**Confidence:** 4

**Metareview:**

The paper proposes a novel approach to address class imbalance in medical image classification tasks by integrating class-imbalance-aware loss functions, specifically Matthews correlation coefficient (MCC) and F1 score, with cross-entropy loss. The study demonstrates significant improvements in classification performance, particularly for minority classes, across various medical imaging datasets. While the approach shows promise, reviewers raise concerns about the evaluation methodology and suggest further experiments and analyses to strengthen the findings.

The paper presents a promising approach for addressing class imbalance in medical image classification tasks. To strengthen the findings, reviewers suggest expanding the evaluation to include more target datasets, providing a clearer comparison of different loss function combinations, and conducting additional analyses to enhance the robustness and generalizability of the approach. Addressing these concerns would further solidify the paper's contribution and impact in the field.

---

### Decision · Program_Chairs · 2024-04-06

Accept (Poster)